# Self-care interventions to assist family physicians with mental health care of older patients during the COVID-19 pandemic: Feasibility, acceptability, and outcomes of a pilot randomized controlled trial

**Mark J. Yaffe**[1,2,3]*, **Jane McCusker**[3,4], **Sylvie D. Lambert**[3,5], **Jeannie Haggerty**[1,3], **Ari N. Meguerditchian**[3,6], **Marc Pineault**[7], **Alexandra Barnabé**[8], **Eric Belzile**[3], **Simona Minotti**[3,9,10], **Manon de Raad**[3]

1 Department of Family Medicine, McGill University, Montreal, Quebec, Canada, 2 St. Mary's Hospital Department of Family Medicine, Montreal, Quebec, Canada, 3 St. Mary's Research Centre, Montreal, Quebec, Canada, 4 Department of Epidemiology and Biostatistics, McGill University, Montreal, Quebec, Canada, 5 Ingram School of Nursing, McGill University, Montreal, Quebec, Canada, 6 Departments of Surgery and Oncology, McGill University, Montreal, Quebec, Canada, 7 Plakett Clinical Services, Montreal, Quebec, Canada, 8 Department of Psychology, McGill University, Montreal, Quebec, Canada, 9 Institute for Better Health, Trillium Health Partners, Mississauga, Ontario, Canada, 10 Biostatistics Division, Dalla Lana School of Public Health, University of Toronto, Toronto, Ontario, Canada

* mark.yaffe@mcgill.ca

**Data Availability Statement:** Data access restriction Consent for data sharing through data

## Abstract

### Background

The COVID-19 pandemic has required family physicians to rapidly address increasing mental health problems with limited resources. Vulnerable home-based seniors with chronic physical conditions and commonly undermanaged symptoms of anxiety and depression were recruited in this pilot study to compare two brief self-care intervention strategies for the management of symptoms of depression and/or anxiety.

### Methods

We conducted a pilot RCT to compare two tele-health strategies to address mental health symptoms either with 1) validated CBT self-care tools plus up to three telephone calls from a trained lay coach vs. 2) the CBT self-guided tools alone. The interventions were abbreviated from those previously trialed by our team, to enable their completion in 2 months. Objectives were to assess the feasibility of delivering the interventions during a pandemic (recruitment and retention); and assess the comparative acceptability of the interventions across the two groups (satisfaction and tool use); and estimate preliminary comparative effectiveness of the interventions on severity of depression and anxiety symptoms. Because we were interested in whether the interventions were acceptable to a wide range of older adults, no mental health screening for eligibility was performed.

repositories was not sought from study participants as part of the informed consent process. As such, the evaluating Research Ethics Committee requests that any request for access to data be made to the corresponding author or Research Ethics Committee directly, to be considered on a case-by-case basis. The Research Ethics Committee can be reached at recherche.comtl@ssss.gouv.qc.ca.

**Funding:** Funding received by MY from: Foundation for Advancing Family Medicine of the College of Family Physicians of Canada and Foundation of Canadian Medical Association COVID-19 Pandemic Response & Impact Grant Program (Co-RIG) - no funding number https://fafm.cfpc.ca/corig/ The funders had no role in study design, data collection and analysis, decision to publish, or preparation of the manuscript.

**Competing interests:** The authors have declared that no competing interests exist.

**Abbreviations:** CBT, Cognitive Behaviour Therapy; RCT, Randomized Controlled Tria; RAs, : Research assistants; BOMC, Blessed Orientation-Memory-Concentration questionnaire; PHQ-9, Patient Health Questionnaire 9 item; GAD-7, Generalized Anxiety Disorder questionnaire 7 item; CSQ-3, 3 item Client Satisfaction Questionnaire; CLSA, Canadian Longitudinal Study on Aging.

## Results

90 eligible patients were randomized. 93% of study completers consulted the self-care tools and 84% of those in the coached arm received at least some coaching support. Satisfaction scores were high among participants in both groups. No difference in depression and anxiety outcomes between the coached and non-coached participants was observed, but coaching was found to have a significant effect on participants' use and perceived helpfulness of the tools.

## Conclusion

Both interventions were feasible and acceptable to patients. Trained lay coaching increased patients' engagement with the tools. Self-care tools offer a low cost and acceptable remote activity that can be targeted to those with immediate needs. While effectiveness results were inconclusive, this may be due to the lack of eligibility screening for mental health symptoms, abbreviated toolkit, and fewer coaching sessions than those used in our previous effective interventions.

## Trial registration

ClinicalTrials.gov Identifier: NCT0460937.

## Background

Depression is the leading cause of disability worldwide [1]. A cross-sectional analysis of depression and anxiety symptoms during the COVID-19 pandemic among Canadians aged 55 + found 26% reporting depression symptoms and 24% anxiety symptoms [2]. Other studies have shown elevated levels of anxiety and depression in the initial phase of the COVID-19 pandemic with psychological distress persisting into subsequent weeks [3]. Such mental health problems are generally several-fold more prevalent in those with chronic physical conditions in comparison to counterparts without [4–6]. Containment measures during a pandemic may also exacerbate pre-existing problems or generate new ones [7–10]; this can be especially true for older, home-based adults who find themselves more isolated [11]. Scalable, low-cost mental health interventions for older adults living with chronic physical conditions must therefore be identified and developed.

Self-care interventions are recommended as the first level in a stepped care program for managing and treating mild-to-moderate depression and anxiety [12, 13]. Meta-analyses affirm that guided self-care interventions, which use principles of Cognitive Behavioral Therapy (CBT) for depression and anxiety have comparable effects to face-to-face psychological therapies [14, 15], and that guided CBT self-care is more effective than non-guided [16]. CBT interventions are well-validated for depression and anxiety which can promote an increased sense of control over thoughts and behaviors [15]. This could be especially beneficial during an unpredictable and uncontrollable situation like a pandemic [17, 18]. When face–to–face contact is challenging, telephone follow-ups have been shown to be a reliable care modality [19], and an effective way to deliver CBT [20].

The COVID-19 pandemic brought about rapid deployment of telephone-based interventions for better access to mental health care for older adults with chronic physical conditions

[21]. Older adults generally prefer telephone to on-line platforms because of age-related cognitive and motor function impairments [22, 23]. Such telephone-based interventions were found to be low cost, without compromising the positive impact of face-to-face interventions [24]. Our team has previously conducted two successful randomized controlled trials (RCTs) of telephone-supported mental health self-care interventions comprising a CBT skills-building toolkit and lay telephone coaching [25, 26]. When compared to self-directed use of the toolkit, telephone guidance from a trained lay coach improved adherence [27] and clinical outcomes in middle-aged and older adults with chronic physical conditions and comorbid depressive symptoms (DIRECT-sc) [25, 28]. Among cancer survivors with depression, frequently accompanied by anxiety, a similar guided self-care intervention (CanDIRECT), improved depression and anxiety symptoms as well as mental health-related quality of life at six months [26].

The pandemic suggested a need to rapidly and differently address mental health problems when resources were limited. Our team evaluated delivery of an abbreviated version of our aforementioned effective CBT self-care toolkit over a time frame one third as long as that previously successful. This intervention aimed for operationalization during the COVID-19 pandemic on a cohort of home-based older adults (65+ years) with chronic physical illness. Because we wished to determine whether the toolkit with or without coaching would be acceptable to a broader audience and anticipated that mental health symptoms would be common among older adults during the pandemic, we opted not to screen for the presence of mental health symptoms as we had in previous studies.

## Study goal and objectives

During an acute stage of an early wave of the COVID-19 pandemic, we set out to conduct Pan-DIRECT, a pilot RCT, to compare two brief self-care intervention strategies for management of symptoms of depression and anxiety. The first involved self-directed use of CBT self-care tools supplied by the study, while the second employed the same tools, but supported by a telephone-based, trained lay coach. Objectives were to assess 1) *feasibility* of delivering the interventions during a pandemic based on: a) successful recruitment of eligible participants into the study, b) participant completion of the interventions, and c) fidelity of delivery of the coaching; 2) comparative *acceptability* of the interventions in the two groups based on degree of use of the self-care tools and satisfaction with the assigned intervention; 3) *preliminary comparative effectiveness* of coached vs non-coached interventions on severity of depression and anxiety symptoms at eight weeks; and 4) participating patients' family physicians' views on the value and acceptability of study-generated information sent to them about their patients.

## Methods

### Study design and timeline

A single blind, individually randomized, pragmatic pilot RCT of the two self-care strategies was conducted during the SARS-COVID-19 pandemic (local onset March 2020) between October 2020 and April 2021. The strategies tested were 1) selected components of previously validated self-care tools (internet and paper-based), with guidance from up to three coach telephone calls over eight weeks; and 2) use of these self-care tools alone. The trial design adhered to the CONSORT criteria [29], the CONSORT extension for pragmatic trials [30], and guidelines for behavioral trials [31]. The protocol was registered on clinicaltrials.gov (Identifier: NCT04609371) on 30/10/2020, and was approved by the hospital research ethics committee (protocol # SMHC 20–10).

## Sample and eligibility criteria

Our study population was obtained from a cohort of 235 patients 65+ years old previously enrolled in a study our team conducted on individuals admitted for physical health problems to either of two acute care hospitals prior to the COVID-19 pandemic [32]. On discharge they were invited to consent to be contacted again for possible participation in other research projects.

Eligibility for the PanDIRECT study included being age 65 or older, English or French speaking, and autonomous home dwelling. People living in long-term care or other medicalized facilities were excluded, but those receiving medical or social services in their own homes were not. Other exclusion criteria were: moderate to severe cognitive impairment (measured by a validated telephone screening interview, see below), inability to read in English or French (self-reported), hearing impairment (judged by research staff over the telephone), self-reported ongoing counseling or psychological therapy begun prior to recruitment (since such treatments might conflict with the self-care interventions [28]). Participants who started therapy after enrolment were not removed from the study. Those expressing suicidal intent (assessed during the screening interview) received further assessment / possible emergency referral and exclusion from the study. Contrary to our previous studies of mental health self-care interventions, we did not limit enrolment to individuals with at least mild depression or anxiety symptoms. Asymptomatic individuals who were interested in the intervention and were otherwise eligible were included in the study sample since tool use might improve overall coping skills during the pandemic, and possibly limit development or worsening of symptoms over a longer time frame [33].

## Contact and consent

Research assistants (RAs) attempted telephone contact with the 235 individuals to introduce the study, establish interest, conduct a brief screening interview to assess eligibility and obtain verbal consent following description of study goals and involvement. Consent was documented by the research assistant and calls were audio-recorded to preserve the consent process. A copy of the consent form was postal mailed or sent electronically to respondents.

## Randomization

On study entry, participants were randomized to the self-directed arm (tools only) or the guided arm (tools with coaching) by a computer-generated randomization schedule using random block sizes, with an allocation ratio of 1:1 (using SAS version 9.4). The sequence was concealed and a participant's assignment appeared only once the coordinator entered the unique participant ID, date of enrolment and information on symptom severity. Participants were stratified based on whether their depression and/or anxiety symptoms were none to mild (as measured by the 9-item Patient Health Questionnaire (PHQ-9) and/or 7-item Generalized Anxiety Disorder questionnaire (GAD-7), scores <10) or moderate to severe (PHQ-9 and/or GAD-7 scores $\geq$10). A discussion of the psychometrics of these tools follows.

## Measures used, data collected

In the eligibility screening interviews RAs administered the Blessed Orientation-Memory-Concentration (BOMC) test, a reliable (retest reliability coefficient of 0.88) and valid (internal consistency: Cronbach's alpha coefficient of 0.89; construct validity as compared to the Mini-Mental State Examination: correlation coefficient of 0.84) six item cognitive screen to assess possible cognitive impairment (with scores of eligible, consenting patients retained as study

data for analyses) [34]. There were also questions on living arrangements, therapy, and ability to read English or French. The suicidal ideation item from the PHQ-9 was used to identify possible suicidal intent.

Among eligible and consenting participants, PHQ-9 results were used as continuous and categorical measures of severity of depressive symptoms [35]. This tool was selected because of its wide use in medical populations, and sensitivity to change [36]. The PHQ-9 has demonstrated good internal consistency across various populations (Cronbach's alpha ranging from 0.79 to 0.89); high test-retest correlations, with coefficients ranging from 0.84 to 0.96 over periods ranging from 1 to 8 weeks; and good construct validity, showing strong correlations with other depression measures and diagnostic criteria for depression [35, 36]. Scores range from 0–27 with established severity ranges (mild: 5–9; moderate: 10–14; moderately severe: 15–19; severe: 20+), and criterion for diagnosis of clinically significant depression being 10 [37].

The 7-item GAD-7 questionnaire was similarly used for continuous and categorical measures of severity of anxiety symptoms [38], with scores ranging from 0–21, with established severity ranges (mild: 5–9; moderate: 10–14; severe: 15+), and criterion for diagnosis of clinically-significant anxiety being 10. The GAD-7 has demonstrated good sensitivity and specificity, a Cronbach's alpha ranging from 0.83 to 0.93 and high positive correlation with other measures of anxiety [39].

A 4-item CAGE alcohol abuse screening questionnaire (score ranging from 0–4) was used for descriptive purposes of possible co-morbid alcohol abuse or dependency [40]. Although a score of 2 or more is considered clinically significant, a score of 1 or more results in greater sensitivity and has been used in some studies [41]. The screening tool has been compared with the Diagnostic Interview Schedule (DIS) and found to have a sensitivity of 92% and a specificity of 82% for identifying alcohol dependence [42].

The baseline interview also included questions on hospitalizations, emergency department visits, counselling in the previous six months, COVID diagnoses in participants or family members, presence of a family caregiver, and reception of homecare services. To reduce interview burden, data collected in the previous study (age, sex, country of origin, and education level) were extracted.

After the eight-week intervention, an RA telephoned participants for a structured follow-up interview. All baseline measures (except the CAGE questionnaire) were re-administered, along with questions about experience in the study: use of the self-care materials, perceived helpfulness in managing mental health problems (on a four point scale, from "not at all helpful" to "very helpful"), use of any health services including counseling or therapy started during the study period, and satisfaction with the intervention (with a three item version of the Client Satisfaction Questionnaire (CSQ-3) [43, 44]). The CSQ has demonstrated internal consistency, with reported alpha coefficients ranging from 0.80 to 0.93 [45, 46]. It correlates with other measures of client satisfaction, treatment outcome, and therapist ratings [45]. *Stage of tool use* was assessed through three categories: 'didn't use', 'just started', and 'well underway', inspired by the stage of change measure validated by Sarkin et al. (2001) [47]. Participants reporting at least some material use were queried to assess *level of use* (level 1 was limited to reading contents of the materials; level 2 to having some engagement with materials to identify relevant self-care approaches; and level 3 indicating that some approaches were at least somewhat applied) [48]. We cross-validated these self-reported tool use reports with information recorded in the coach logs (described below).

Participants were invited to provide their family physicians' contact information so that the latter could be mailed a report of their patients' study involvement once the patients had completed the study. The reports included brief information on the study, participants' PHQ-9 and GAD-7 scores at baseline and follow-up, instructions on how to interpret the scores, and a brief postal survey (for return by pre-paid stamped/addressed envelope, fax, or email). This

survey enquired about (1) usefulness of the report; (2) their general familiarity with PHQ-9 and GAD-7 screening tools; and (3) their interactions with participants during the study period. Reminders were sent to non-responders four weeks after the first mailing.

## Interventions

A mailing to participants' homes using a premium guaranteed delivery service contained a study welcome letter, self-use materials, description of what to expect next (coach calls, if relevant), and expected timeline for a follow-up questionnaire. The self-care CBT-based toolkit was adapted from tools used in our previous studies [25, 26, 49] to accommodate the shorter intervention period. The main changes were: no initial screen; making toolkit available to those without clinically significant depression or anxiety symptoms; limiting the number of tools sent to two (versus the entire toolkit of over eight tools); reduction in the length of certain tools (i.e. providing one chapter of a workbook as opposed to the whole workbook); and an abbreviated guidance schedule of up to three calls over eight weeks (compared to up to 15 calls over six months).

In this study, specific self-care tools were assigned to each participant according to a study-developed algorithm for particular positive findings identified on the PHQ-9 and GAD-7 at baseline. Each participant was therefore assigned one CBT primary tool, either the Reactivating Your Life chapter from the Antidepressant Skills Workbook [50, 51] or the Managing Worry chapter from the Positive Coping With Health Conditions Workbook [52]; and one secondary tool, either a mood monitoring tool, a relaxation audio tool, information on exercise and healthy eating, information on sleep [50] or summaries on emotional eating (Toolkit described in S1 Appendix).

Guidance on the use of the materials in the intervention arm was delivered by individuals who had recently obtained Bachelor's degrees in psychology and whose training and activities were supervised by a coach supervisor. The latter, a registered psychotherapist and clinical psychology PhD candidate, was familiar with the interventions used in the team's previous RCTs, had worked previously with the co-investigators, and had led the creation of the algorithm in the current study that personalized the selection of tools for each participant. Each of the intervention arm participants was followed by one of three study coaches who were bilingual (English, French), had strong organizational and interpersonal skills, but no formal training in CBT. They received a half day training via videoconference, followed by mock practice sessions over the telephone. Initial calls with participants took place one week following toolkit deliveries, with a mandate of a maximum three calls over eight weeks and a recommended 15 to 20 minutes per call. Coach activities were guided by a Coach Manual adapted from that used in our previous two RCTs [25, 26]. They had no role in psychotherapy or counselling, and their function was limited to assisting participants by encouraging use of assigned tools and helping to formulate goals to meet objectives related to the strategies described in the tools.

The coaches maintained logs of all telephone contacts attempted or completed, including their duration. They noted which tools were recommended and used, as well as participants' levels of tool use using a 3-point scale: 1) coach introduced tool, 2) participant read tool, and 3) participant implemented strategies in the tool or tried the exercises at each of the calls. The coach supervisor reviewed the logs, along with random audio taped telephone contacts, employing a fidelity checklist (adapted from our previous research [25, 26]), for assessing coaching done in accordance with the protocol, and for verifying coaching data in the logs addressing the feasibility objectives.

## Data analysis

Data analysis was conducted according to the CONSORT guidelines [29, 30]. All the quantitative analyses were carried out with SAS version 9.4 and STATA 15.0 software.

Baseline imbalance was assessed by computing the standardized difference [53] between study arms (coached and self-directed) [54] with respect to six pre-specified baseline variables (age, sex, hospital recruited from in the original study from which the cohort was created, COVID-19 diagnosis, PHQ-9, GAD-7), and nine other variables selected by the research team (see Table 1). Any variable with a standardized difference ≥ 0.15 was considered evidence of imbalance [55].

The feasibility of the intervention was assessed using descriptive statistics to describe refusal and uptake rates, completion of surveys, and the rates of missing data. Analyses including only participants who completed the eight-week follow-up were performed. The standardized difference and 95% confidence interval was computed to compare the study arms with respect to stage and level of self-care tool use and the satisfaction [53, 56]. Unadjusted effect sizes (Cohen d) defined as the study group mean difference in the outcome score, divided by the pooled standard deviation [57] and 95% confidence intervals [56, 58] were computed for both primary outcomes. In addition, a linear regression model was fitted to adjust for baseline imbalance [59], while adjusted effect sizes and 95% confidence intervals (95% CI) were computed as the Beta estimate of the intervention group divided by the pooled standard deviation obtained from the unadjusted analysis. For the regression analyses, standard model checking techniques were applied (e.g.: residual plots) [59]. Finally, the standardized difference was also used to compare characteristics of participants who completed the eight week follow-up survey with participants who did not (see Table A in S2 Appendix). The analysis by intention-to-treat was performed to account for potential selection bias due to differential losses to follow up; missing data were imputed using Multiple Imputation by Chained Equations [60, 61], see S2 Appendix for more details.

We cross-validated the self-reported stage of use (for primary and secondary tools) and level of use (for primary tools only, as participants only self-reported level of use for the workbooks) variables against tool use information from coach logs, using the Cramer-V statistic [62]; this statistic ranges from 0 to 1 and can be interpreted as a) weak: 0.0 to 0.10, b) moderate: >0.10 to 0.15, c) strong: >0.15 to 0.25, and d) very strong >0.25 [63].

## Sample size

We followed guidelines for minimum sample sizes for pilot studies [64]. With a final sample size of over 40 participants in each group, we were well above the recommendations.

## Results

### Feasibility: Recruitment and study sample

Of 235 older adults in our previous study consenting to follow-up [32], 75 were not reachable by telephone due to numbers no longer in service, patients having died, calls not answered or messages not returned. Fig 1 summarizes the outcome of successful contact of 160 individuals. Forty-four were not interested, with over half indicating they did not have time or not providing a reason. Twenty-three did not meet eligibility criteria (mostly due to either visual, hearing or cognitive impairment). Of the remaining 93 who engaged in the current study's consent process, 90 consented to participation.

Characteristics of these 90 older adults (see Table 1) include a mean age (standard deviation) of 79.4 (6.8), slightly more females, 2/3 with at least post-secondary education, high

**Table 1. Assessing baseline imbalance variables across study group (n = 90).**

| Variables | Overall (n = 90) | | Coached (n = 44) | | Self-directed (n = 46) | | Standardized difference |
|---|---|---|---|---|---|---|---|
| **Baseline variables selected a priori (listed in protocol):** | | | | | | | |
| PHQ-9, n (%) | | | | | | | **0.27** |
| 0–4 | 52 | 57.8 | 26 | 59.1 | 26 | 56.5 | |
| 5–9 | 19 | 21.1 | 10 | 22.7 | 9 | 19.6 | |
| 10–14 | 10 | 11.1 | 3 | 6.8 | 7 | 15.2 | |
| 15+ | 9 | 10.0 | 5 | 11.4 | 4 | 8.7 | |
| PHQ-9, mean (SD) | 5.8 (5.3) | | 5.8 (5.4) | | 5.8 (5.2) | | 0.00 |
| GAD-7, n (%) | | | | | | | **0.34** |
| 0–4 | 65 | 72.2 | 32 | 72.7 | 33 | 71.7 | |
| 5–9 | 18 | 20.0 | 7 | 15.9 | 11 | 23.9 | |
| 10–14 | 6 | 6.7 | 4 | 9.1 | 2 | 4.4 | |
| 15+ | 1 | 1.1 | 1 | 2.3 | 0 | 0.0 | |
| GAD-7, mean (SD) | 3.3 (3.9) | | 3.5 (4.3) | | 3.2 (3.4) | | 0.09 |
| Age, n (%) | | | | | | | **0.42** |
| 65–74 | 24 | 26.7 | 12 | 27.3 | 12 | 26.1 | |
| 75–84 | 44 | 48.9 | 25 | 56.8 | 19 | 41.3 | |
| 85+ | 22 | 24.4 | 7 | 15.9 | 15 | 32.6 | |
| Age, mean (SD) | 79.4 (6.8) | | 79.0 (5.8) | | 79.7 (7.7) | | 0.11 |
| Female, n % | 51 | 56.7 | 27 | 61.4 | 24 | 52.2 | **0.19** |
| Hospital recruited from in original study, n (%) | | | | | | | 0.08 |
| Hospital A | 37 | 41.1 | 19 | 43.2 | 18 | 39.1 | |
| Hospital B | 53 | 58.9 | 25 | 56.8 | 28 | 60.9 | |
| **Baseline variables NOT selected a priori (n (%)):** | | | | | | | |
| Born in Canada | 58 | 64.4 | 29 | 65.9 | 29 | 63.0 | 0.06 |
| Cognitive impairment (BOMC) | | | | | | | 0.02 |
| [0–4] normal cognition | 70 | 77.8 | 34 | 77.3 | 36 | 78.3 | |
| [5–9] questionable | 20 | 22.2 | 10 | 22.7 | 10 | 21.7 | |
| Education (from PCAP baseline) | | | | | | | **0.59** |
| University degree | 40 | 44.4 | 18 | 40.9 | 22 | 47.8 | |
| High school and post | 20 | 22.2 | 13 | 29.6 | 7 | 15.2 | |
| Completed high school | 18 | 20.0 | 5 | 11.4 | 13 | 28.3 | |
| Less than high school | 12 | 13.3 | 8 | 18.2 | 4 | 8.7 | |
| Presence of a caregiver | | | | | | | **0.48** |
| No | 34 | 37.8 | 21 | 47.7 | 13 | 28.3 | |
| Yes-does not live with | 32 | 35.6 | 15 | 34.1 | 17 | 37.0 | |
| Yes-live with | 24 | 26.7 | 8 | 18.2 | 16 | 34.8 | |
| CAGE score | | | | | | | **0.42** |
| 0 | 78 | 91.8 | 40 | 97.6 | 38 | 86.4 | |
| 1+ | 7 | 8.2 | 1 | 2.4 | 6 | 13.6 | |
| (missing) | (5) | | (3) | | (2) | | |
| Receiving homecare services from the CLSC | | | | | | | 0.02 |
| No | 70 | 80.5 | 34 | 81.0 | 36 | 80.0 | |
| Yes | 17 | 19.5 | 8 | 19.0 | 9 | 20.0 | |
| (missing) | (3) | | (2) | | (1) | | |
| Previous counseling since March 2020 | | | | | | | **0.53** |
| No | 79 | 94.0 | 35 | 87.5 | 44 | 100.0 | |
| Yes | 5 | 6.0 | 5 | 12.5 | 0 | 0.0 | |

(*Continued*)

**Table 1.** (Continued)

| Variables | Overall (n = 90) | | Coached (n = 44) | | Self-directed (n = 46) | | Standardized difference |
|---|---|---|---|---|---|---|---|
| (missing) | (6) | | (4) | | (2) | | |
| ED visits without hospitalization* | | | | | | | 0.13 |
| No | 78 | 88.6 | 39 | 90.7 | 39 | 86.7 | |
| Yes | 10 | 11.4 | 4 | 9.3 | 6 | 13.3 | |
| (missing) | (2) | | (1) | | (1) | | |
| Hospitalization* | | | | | | | **0.24** |
| No | 66 | 75.0 | 30 | 69.8 | 36 | 80.0 | |
| Yes | 22 | 25.0 | 13 | 30.2 | 9 | 20.0 | |
| (missing) | (2) | | (1) | | (1) | | |

Any Standardized Differences (SD) were identified in bold font when SD was greater than 0.15, for each variable this difference correspond to a clinically significant baseline imbalance across study groups; *unrelated to COVID

cognitive functioning (as determined by the BOMC at screening), 1 having had a positive COVID-19 diagnosis, 1/3 having been to an ER during the COVID pandemic, and 2/3 having a family caregiver. Since the pandemic's onset, 31.4% perceived experiencing worse physical health, and 47.2% worse mental health (not indicated in table), while at the time of the study PHQ-9 and GAD-7 scores identified no or only mild symptoms of depression or anxiety.

The 90 were randomized to the coached group (n = 44), or the self-directed group (n = 46). Table 1 identifies nine variables that were imbalanced between the two groups. Compared to the self-directed group, the following were more frequent in the coached group: scores in the less depressed or anxious categories (although mean scores were similar); less than 85 years of age; female; completed less than a university education; no caregiver; a CAGE score of 0; previous counseling; and hospitalization or emergency room visit during the previous six months. One participant (in the control group) started therapy after enrolment. 68.9% (62 /90) completed the follow-up questionnaire, comprising 63.6% (28/44) of the coached cohort and 73.9% (34/46) of the self-directed cohort.

## Feasibility: Intervention delivery

All randomized participants received their tools, except one in the self-directed group (multiple attempts at delivery were unsuccessful). In the coached group, 37/44 received at least one coach call (84%), 52% completed all three calls (Table 2). Among the seven who did not receive at least one call, five could not be reached despite multiple attempts and two declined the coach calls indicating they were not interested. The average call length was 15.0 minutes, generally occurring at two week intervals. Participants who did not complete follow-up (n = 16) received fewer calls than completers of follow-up (mean 1.2 vs 2.5 calls, respectively).

At follow-up, over 80% of participants who had received at least one coach call reported that the number of calls felt appropriate to them and had helped them understand and use the tools. Only 32% indicated that they would have preferred to use the tools without coaching support.

Twenty-eight coach calls were randomly selected for fidelity review. On average, 96% of the coaching components were delivered as per the intervention protocol (see Table 2).

## Acceptability of intervention

Table 3 describes participants' tool use (the participant in the self-directed groups who did not receive the tools is excluded). We noted a significant effect of coaching on *stage of use* of both

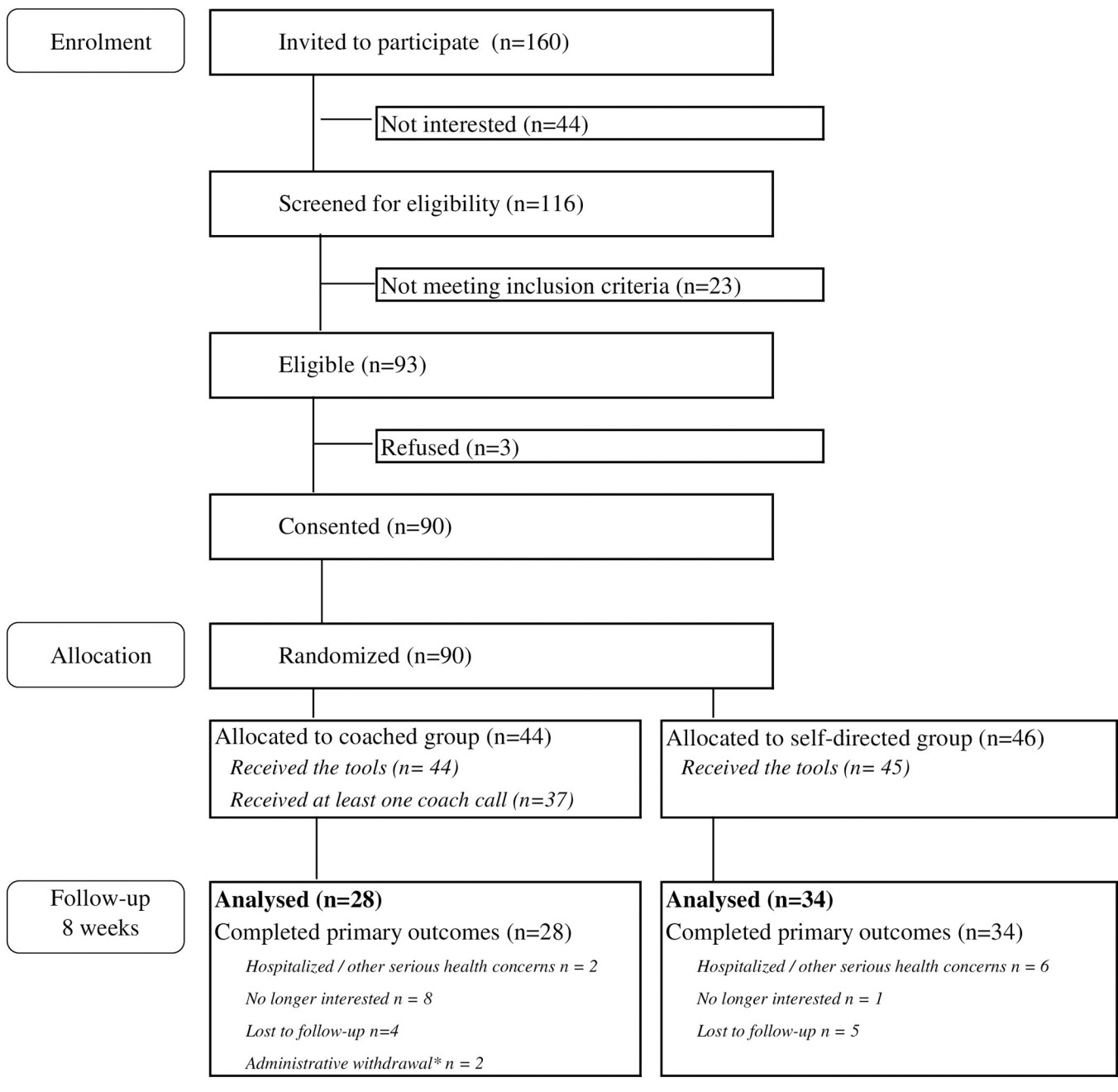

**Fig 1. Flowchart.**

the workbook and secondary tool assigned, as well as on perceived helpfulness of the secondary tool. Coaching was not associated with *level of use* among users or with satisfaction about the intervention, which was generally high in both groups.

An exploration of the association between coach-recorded tool use and participants' self-reported tool use was examined. The Cramer-V statistic computed by comparing self-reported vs coach reported stage of use was 0.47 (n = 27), while for level of use it was 0.39 (n = 27). Both of these are considered to be very strong associations [63].

**Table 2. Feasibility of intervention delivery.**

| Variables | Overall | Completers of follow-up | Non-completers of follow-up |
|---|---|---|---|
| **Coach log:** | **(n = 44)** | **(n = 28)** | **(n = 16)** |
| Coach calls completed, n (%) | | | |
| 0 | 7 (15.9) | 4 (14.3) | 3 (18.8) |
| 1 | 9 (20.5) | 0 (0.0) | 9 (56.3) |
| 2 | 5 (11.4) | 3 (10.7) | 2 (12.5) |
| 3 | 23 (52.3) | 21 (75.0) | 2 (12.5) |
| mean (SD) | 2.0 (1.2) | 2.5 (1.1) | 1.2 (0.9) |
| **Among users (1+ coach calls)** | **(n = 37)** | **(n = 24)** | **(n = 13)** |
| Average duration (min) per coach call | | | |
| mean (SD) | 15.0 (6.6) | 17.7 (4.4) | 9.9 (7.1) |
| median [Q1-Q3] | 17 [10.5–19.7] | 18.5 [15–20.2] | 10.3 [3–17] |
| Cumulative coach contact (min) per participant | | | |
| mean (SD) | 39.1 (22.1) | 51.0 (13.0) | 17.1 (18.3) |
| median [Q1-Q3] | 46 [18–59] | 55.5 [45; 60] | 13 [3; 21] |
| Coach reported stage of use of primary tool (workbook) | | | |
| No use reported | 14 (37.8) | 3 (12.5) | 11 (84.6) |
| Use reported at only 1 coach call (just started) | 6 (16.2) | 4 (16.7) | 2 (15.4) |
| Use reported at at least 2 calls (well underway) | 17 (45.9) | 17 (70.8) | 0 (0.0) |
| Coach reported stage of use of secondary tool | | | |
| No use reported | 27 (73.0) | 16 (66.7) | 11 (84.6) |
| Use reported at only 1 coach call (just started) | 7 (18.9) | 5 (20.8) | 2 (15.4) |
| Use reported at at least 2 calls (well underway) | 3 (8.1) | 3 (12.5) | 0 (0.0) |
| Coach reported level of use of primary tool (workbook) | | | |
| No introduction reported | 9 (24.3) | 1 (4.2) | 8 (61.5) |
| Tool introduced only (level 1) | 5 (13.5) | 2 (8.3) | 3 (23.1) |
| Tool read by participant (level 2) | 12 (32.4) | 10 (41.7) | 2 (15.4) |
| Participant implemented strategies from the tool (level 3) | 11 (29.8) | 11 (45.8) | 0 (0.0) |
| **Fidelity:** (a random selection of coach calls) | **(n = 28)** | | |
| Procedure score (0–100), mean (SD) | 95.0 (9.7) | | |
| Contents score (0–100), mean (SD) | 97.0 (7.8) | | |
| **Feedback on the coach calls** | | **(n = 22)** | |
| (at follow-up section completed and among those with at least 1 coach call): | | | |
| Number of calls n (%) | | | |
| Too many | | 1 (4.5) | |
| Too few | | 2 (9.1) | |
| The right number | | 19 (86.4) | |
| Coach helped to understand and use the self-care tools n (%) | | | |
| Yes | | 18 (81.8) | |
| Would have preferred to use tools without coaching support n (%) | | | |
| Yes | | 7 (31.8) | |

## Potential effectiveness of the coaching intervention

Table 4 summarizes the effects of the eight week coaching intervention on PHQ-9 and GAD-7. Effect sizes were small (less than 0.2) in unadjusted and adjusted analyses. Descriptive results using depression and anxiety categories do not suggest important differences between the

**Table 3. Comparative acceptability of the two interventions (n = 61).**

| | Coached | | Self-directed | | |
|---|---|---|---|---|---|
| | **(n = 28)** | | **(n = 33)** | | **Standardized difference[95% CI]** |
| **Variables** | **n** | **%** | **n** | **%** | |
| **Workbook assigned** | | | | | 0.34 [-0.17; 0.85] |
| Reactivating Your Life Workbook | 21 | 75.0 | 29 | 87.9 | |
| Managing Worry Workbook | 7 | 25.0 | 4 | 12.1 | |
| Stage of use of workbook assigned: | (n = 28) | | (n = 33) | | **0.54 [0.03; 1.05]** |
| No plan to use / didn't look at | 0 | 0.0 | 4 | 12.1 | |
| Just started | 5 | 17.9 | 4 | 12.1 | |
| Well underway | 23 | 82.1 | 25 | 75.8 | |
| Level of use (among users) | (n = 28) | | (n = 29) | | 0.31 [-0.21; 0.83] |
| Level 1: Read the workbook but did not apply it | 6 | 21.4 | 10 | 34.5 | |
| Level 2: Used workbook to identify behaviours to change | 7 | 25.0 | 5 | 17.2 | |
| Level 3: Used workbook to implement goals for behaviour change | 15 | 53.6 | 14 | 48.3 | |
| Helpfulness of workbook (among users) | (n = 28) | | (n = 29) | | 0.37 [-0.15; 0.89] |
| Not at all helpful | 3 | 10.7 | 3 | 10.3 | |
| A little helpful | 3 | 10.7 | 7 | 24.1 | |
| Moderately helpful | 11 | 39.3 | 8 | 27.6 | |
| Very helpful | 11 | 39.3 | 11 | 37.9 | |
| Plan to continue using? (among users) | (n = 28) | | (n = 29) | | 0.40 [-0.12; 0.92] |
| Yes | 20 | 71.4 | 15 | 51.7 | |
| Not sure | 3 | 10.7 | 6 | 20.7 | |
| No | 5 | 17.9 | 8 | 27.6 | |
| **Secondary tool assigned** | | | | | 0.33 [-0.18; 0.84] |
| Mood monitoring | 9 | 32.1 | 9 | 27.3 | |
| Relaxation CD | 9 | 32.1 | 13 | 39.4 | |
| Diet / exercise | 0 | 0.0 | 0 | 0.0 | |
| Emotional eating | 1 | 3.6 | 0 | 0.0 | |
| Sleep | 9 | 32.1 | 11 | 33.3 | |
| Stage of use of secondary tool | (n = 28) | | (n = 33) | | **0.99 [0.46; 1.52]** |
| No plan to use / didn't look at | 2 | 7.1 | 15 | 44.1 | |
| Intend to use | 6 | 21.4 | 2 | 5.9 | |
| Just started | 5 | 17.9 | 5 | 14.7 | |
| Well underway | 15 | 53.6 | 12 | 35.3 | |
| Helpfulness of secondary tool (among users) | (n = 20) | | (n = 17) | | **0.83 [0.14; 1.52]** |
| Not at all helpful | 2 | 10.5 | 3 | 18.8 | |
| A little helpful | 1 | 5.3 | 5 | 31.3 | |
| Moderately helpful | 6 | 31.6 | 3 | 18.8 | |
| Very helpful | 10 | 52.6 | 5 | 31.3 | |
| (missing) | (1) | | (1) | | |
| Plan to continue using? (among users) | (n = 20) | | (n = 17) | | 0.38 [-0.27; 1.03] |
| Yes | 13 | 65.0 | 8 | 47.1 | |
| Not sure | 2 | 10.0 | 2 | 11.8 | |
| No | 5 | 25.0 | 7 | 41.2 | |
| **Client Satisfaction Questionnaire (CSQ):** | | | | | |
| Mean (SD) | 10.0 (1.8) | | 9.3 (1.6) | | 0.39 [-0.14; 0.92] |
| High satisfaction | 24 | 88.9 | 21 | 72.4 | 0.43 [-0.10; 0.96] |
| (missing) | (1) | | (4) | | |

*(Continued)*

**Table 3.** (Continued)

| | Coached | | Self-directed | | |
|---|---|---|---|---|---|
| | (n = 28) | | (n = 33) | | Standardized difference[95% CI] |
| Variables | n | % | n | % | |
| Participant who did not receive the toolkit is excluded | Significant Standardized Differences are in bold font | | | | |

Participant who did not receive the toolkit is excluded. Significant Standardized Differences are in bold font

study groups or between baseline and follow-up. Sensitivity analyses taking into account missing data found similar results (S2 Appendix).

## Feedback from family physicians

69% (43/62) of study completers agreed to have reports sent to their family physicians at the end of the study. 46.5% (20/43) returned completed surveys included with the report. The respondents appeared to be involved in mental health care given that 80% and 75% respectively used the PHQ-9 and the GAD-7. 55% (11/20) found the patient information sent to them useful, while the remainder held a neutral opinion. Nonetheless 70% (14/20) of physicians kept the information for filing in the patients' medical records, 20% (4/20) used it to try to initiate or confirm patient follow-up, and 90% (18/20) indicated interest in future reception of similar patient information, if it was available.

## Discussion

We have reported on the feasibility, acceptability, and comparative preliminary effectiveness of delivering two brief tele-health strategies to address potential symptoms of depression and/ or anxiety: 1) validated self-care tools plus up to three telephone lay coach calls; versus 2) self-guided tools alone. The tools, based on the principles of CBT, have been shown to be effective in previous RCTs by our team [25, 26]. Feasibility was evaluated using measures of recruitment, retention and fidelity of intervention delivery; patient acceptability of the interventions using measures of intervention adherence and satisfaction; and preliminary effectiveness of the interventions on severity of depression and anxiety after eight weeks. Given that this was a pilot RCT, statistical significance (or lack of it) cannot be claimed; however results are encouraging and the test for efficacy (or effectiveness) must be tested in an appropriately powered RCT.

This study offers a methodological improvement that may serve future studies of similar interventions: rather than asking about tool use versus non-use, we asked participants to report their stage of use (three point scale, 'didn't use', 'just started', and 'well underway'), level of use (three levels: reading contents of the materials; having some engagement with materials to identify relevant self-care approaches; and indicating that some approaches were at least somewhat applied), and perceived helpfulness (four point scale, from "not at all helpful" to "very helpful"). We were able to validate some of these variables against coach observations. These variables appeared to better discriminate between the study groups in favor of the coached group.

Recruitment was targeted to a previously identified cohort of older adults with known chronic illness. Decision to recruit from this group was informed by literature suggesting its greater risk for mental health problems and for increased cancellation or avoidance of medical care during the early part of the pandemic [65]. We were also influenced by expectation that it would be difficult to obtain potential participant referrals from physician practices already

**Table 4. Comparative outcomes of the two interventions (n = 62).**

| Information | Coached (n = 28) | Self-directed (n = 34) |
|---|---|---|
| **PHQ-9:** | | |
| Continuous (0–27), mean (SD) | | |
| Baseline | 6.4 (5.6) | 6.5 (5.4) |
| 2-month | 5.3 (5.2) | 6.0 (6.0) |
| Effect size (95% CI) at 2-month | | |
| MI approach* | | |
| Unadjusted | 0.16 [-0.35; 0.67] | |
| Adjusted** | 0.18 [-0.25; 0.62] | |
| 2-month completers | | |
| Unadjusted | 0.12 [-0.38; 0.62] | |
| Adjusted** | 0.13 [-0.23; 0.49] | |
| Categorical, n (%) | | |
| Baseline | | |
| 0–4 | 15 (53.6) | 17 (50.0) |
| 5–9 | 7 (25.0) | 8 (23.5) |
| 10–14 | 2 (7.1) | 5 (14.7) |
| 15+ | 4 (14.3) | 4 (11.8) |
| 2-month | | |
| 0–4 | 15 (53.6) | 17 (50.0) |
| 5–9 | 7 (25.0) | 10 (29.4) |
| 10–14 | 5 (17.9) | 3 (8.8) |
| 15+ | 1 (3.6) | 4 (11.8) |
| **GAD-7:** | | |
| Continuous (0–21), mean (SD) | | |
| Baseline | 3.8 (3.9) | 3.5 (3.6) |
| 2-month | 3.0 (4.6) | 2.9 (3.7) |
| Effect size (95% CI) at 2-month | | |
| MI approach* | | |
| Unadjusted | 0.02 [-0.58; 0.61] | |
| Adjusted** | 0.10 [-0.58; 0.69] | |
| 2-month completers | | |
| Unadjusted | -0.02 [-0.52; 0.48] | |
| Adjusted** | 0.01 [-0.44; 0.45] | |
| Categorical, n (%) | | |
| Baseline | | |
| 0–4 | 20 (71.4) | 23 (67.7) |
| 5–9 | 5 (17.9) | 9 (26.5) |
| 10–14 | 3 (10.7) | 2 (5.9) |
| 15+ | 0 (0.0) | 0 (0.0) |
| 2-month | | |
| 0–4 | 22 (78.6) | 28 (82.4) |
| 5–9 | 3 (10.7) | 3 (8.8) |
| 10–14 | 1 (3.6) | 3 (8.8) |
| 15+ | 2 (7.1) | 0 (0.0) |

* Multiple Imputation (MI) was performed to handle the missing data of PHQ-9 or GAD-7 at 2-month

**Adjusted for baseline imbalance: age group, sex, PHQ-9 and GAD-7 group

disrupted or frequently virtual [66]. Even with the approach that we took, a large number of potential participants were not reachable for recruitment, perhaps outcomes of personal and social upheaval during the COVID-19 pandemic. Nonetheless delivery of the intervention was found to be feasible as all participants but one successfully received their tools, and 84% of participants in the coached arm were successfully reached by their coach. Although we did observe slightly more drop-outs in the coached arm, we hypothesize that this difference was due to those participants achieving more rapid comfort with what they had to learn about the self-care tools. These participants may have been less inclined to continue their participation once their needs were met. Recruitment and retention results indicate that delivery of the proposed interventions within the constraints imposed by a pandemic was feasible.

We had postulated that symptoms of anxiety and depression would be present in our sample as a consequence of lifestyle limitations and social isolation imposed by COVID-19 pandemic restrictions and lockdowns. Indeed a large (n = 24,114) telephone survey conducted by the Canadian Longitudinal Study on Aging (CLSA) during the pandemic showed a 7.4% increase in depression prevalence from study entry three years prior to mid pandemic [67]. Hence, the large proportion of asymptomatic participants in our study based on low PHQ-9 and GAD-7 scores was unexpected. As noted above, 47.2% of our participants reported worse mental health since the pandemic onset, yet they had low cross-sectional PHQ-9 and GAD-7 scores at the time of our study, perhaps coinciding with various societal and emotional shifts seen during the pandemic. Other studies have reported elevated symptoms of anxiety and depression in the early phases of the pandemic [4]. As such, it is possible that when we conducted our study some participants may have had few mental health problems, and consequently did not see reason for study participation or engagement. There may be value therefore in re-conducting our study during different stages of a pandemic to see if there are different mental health care needs. As well, in our previous studies, participants on average, received eight coach calls, and were offered a much broader selection of tools. We should consider if the rapid two-month program in the present study, shortened to maximize coach availability to a future large cohort during a pandemic, contributed to the inconclusive comparative efficacy results.

For another perspective, we sought feedback from family physicians of the participants. While the sample size of respondents was small, the participation rate seems understandable given that doctors were surveyed during Wave 3 of the pandemic when many were either difficult to reach because they were providing virtual care from home, or were seconded to provide institutional care. As well, a systematic review of family doctor participation in research projects suggests that the 46.5% participation rate in our study is actually in the higher range for such activity [68]. Therefore, the generally positive reaction of the physicians to this self-care project is encouraging.

Does our intervention offer an option to other care delivery models that may be overburdened or inefficient? During the pandemic in Toronto, Canada, primary care was delivered virtually in 77.5% of cases, with 90.6% of such cases being for anxiety and depression [69]. It is not clear that such an approach adequately responded to the large demand. Meanwhile, digitalized tools are suggested to be of help in mental health care, but a scoping review that identified mobile interventions and apps found that only a handful were actually accessible or had undergone some scientific review [3]. Our self-care programs may therefore present a viable option for large-scale mental health care.

Despite our promising feasibility and acceptability results, no difference in depression and anxiety outcomes between the coached and non-coached participants was observed in the current time reduced intervention. We included non-validated questions related to participants' perception of their physical and mental quality of life, and results seemed to indicate that

respondents felt their health was better at follow-up. However, these results were not significant and could have been mediated by other factors and therefore no conclusions can be drawn. Hence, the absence of a validated quality of life scale in study measures can be seen as a limitation. The self-care activity, however, was generally acceptable to the target population, with satisfaction scores being high in both groups. Family physicians appeared supportive of this self-care approach to managing depression and anxiety, which is encouraging since close to two thirds of all study completers indicated plans to continue using the tools. For those with low concern for depression and anxiety our tools are designed to improve coping skills that may have preventative mental health benefits during a pandemic or in later life [33]. This is encouraging since it has been suggested that programs incorporating behavioral activation may be useful to address loneliness, a precursor of depression in older adults with chronic conditions [70].

## Conclusions

Previous research by our team demonstrated effectiveness of using trained lay coaches and CBT tools to treat mild to moderate depression [49]. In the current study, we attempted to respond to the complexity of conducting clinical care under pandemic conditions, notably by removing any threshold mental health score for study eligibility. This resulted in a limited sample of participants experiencing mental health symptoms, which affected ability to detect effect. As well, the lack of comparative effectiveness may be due to other modifications made to the intervention: reduced number of tools offered, and fewer coach calls. Since participants indicated a favorable impression of the tools, either coach supported or non-supported use of these tools may aid patients in handling anxiety and/or depression symptoms during and after pandemic waves, while preventatively improving their overall coping skills. Telephone-based coaching offers potential added benefit by increasing participant engagement with the tools. Future studies may want to further explore how adherence, using the stage and level of use measures proposed, could impact health outcomes, including cognitive functioning. Since pandemics may mandate social distancing, the intervention is a low cost and acceptable remote activity that can be targeted to those with immediate needs.

## Supporting information

**S1 Checklist. Checklist of items for reporting pragmatic trials.**
(PDF)

**S1 Appendix. Description of the PanDIRECT self-care tools.**
(PDF)

**S2 Appendix. Multiple imputation approach and additional analyses on patient outcomes.**
(PDF)

**S1 Protocol.**
(PDF)

## Author Contributions

**Conceptualization:** Mark J. Yaffe, Jane McCusker, Alexandra Barnabé, Eric Belzile.

**Data curation:** Jane McCusker, Eric Belzile, Manon de Raad.

**Formal analysis:** Mark J. Yaffe, Jane McCusker, Eric Belzile, Simona Minotti, Manon de Raad.

**Funding acquisition:** Mark J. Yaffe, Jane McCusker.

**Investigation:** Mark J. Yaffe, Jane McCusker, Sylvie D. Lambert, Jeannie Haggerty, Ari N. Meguerditchian.

**Methodology:** Mark J. Yaffe, Jane McCusker, Jeannie Haggerty, Eric Belzile, Simona Minotti.

**Project administration:** Alexandra Barnabé, Manon de Raad.

**Resources:** Marc Pineault.

**Supervision:** Mark J. Yaffe, Jane McCusker, Alexandra Barnabé, Manon de Raad.

**Validation:** Mark J. Yaffe, Jane McCusker, Alexandra Barnabé, Manon de Raad.

**Writing – original draft:** Mark J. Yaffe, Jane McCusker, Sylvie D. Lambert, Alexandra Barnabé, Eric Belzile, Manon de Raad.

**Writing – review & editing:** Mark J. Yaffe, Jane McCusker, Jeannie Haggerty, Ari N. Meguerditchian, Marc Pineault, Eric Belzile, Simona Minotti, Manon de Raad.

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
