## [Decision Letter · Decision Letter 0]

3 May 2023

PONE-D-23-02036Self-care interventions to assist family physicians with mental health care of older patients during the COVID-19 pandemic: Feasibility, acceptability, and outcomes of a pilot randomized controlled trialPLOS ONE

Dear Dr. de Raad,

Thank you for submitting your manuscript to PLOS ONE. After careful consideration, we feel that it has merit but does not fully meet PLOS ONE’s publication criteria as it currently stands. Therefore, we invite you to submit a revised version of the manuscript that addresses the points raised during the review process.

Your manuscript has been evaluated by two reviewers, and their comments are available below. While overall positive towards publication, the reviewers have provided comments regarding ways in which the reporting of the study methodology and results could be improved, in addition to other general opportunities for improvement to the manuscript. Please ensure you address each of the reviewers' comments when revising your manuscript.

We look forward to receiving your revised manuscript.

Kind regards,

Hugh Cowley

Staff Editor

PLOS ONE

Journal Requirements:

2. In your statement, please include the full name of the IRB or ethics committee who approved or waived your study, as well as whether or not you obtained informed written or verbal consent. If consent was waived for your study, please include this information in your statement as well.

Reviewers' comments:

Reviewer's Responses to Questions

**Comments to the Author**

1. Is the manuscript technically sound, and do the data support the conclusions?

Reviewer #1: Yes

Reviewer #2: Yes

2. Has the statistical analysis been performed appropriately and rigorously? 

Reviewer #1: Yes

Reviewer #2: Yes

3. Have the authors made all data underlying the findings in their manuscript fully available?

Reviewer #1: Yes

Reviewer #2: No

4. Is the manuscript presented in an intelligible fashion and written in standard English?

Reviewer #1: Yes

Reviewer #2: Yes

5. Review Comments to the Author

Reviewer #1: This pilot RCT determined how self-guided interventions could serve as an adjunctive treatment for family physicians providing mental healthcare to community older adults with various levels of mental health symptoms during the COVID-19 pandemic. They evaluated two telehealth approaches: one that included minimal support by a lay coach and another that was entirely self-guided. They observed that most participants engaged with the CBT-based tools and that most participants in the coached arm experienced support from the lay coach. Although no between-group effects emerged in targeting depression and anxiety symptoms, coaching positively affected uptake and subjective benefit derived from the tool. Overall, this preregistered pilot RCT contributes meaningfully to the digital mental health intervention literature. The sample size was relatively large for a pilot study, the methods were sound (e.g., excluding those who partook in psychological treatments, concealment of treatment allocation, rigorous supervision, high fidelity, etc.), and the paper was well-written. This reviewer lists the following feedback to improve the article further before publication.

1. Background, lines 2-8: The paragraph should end with a sentence highlighting the importance of identifying and developing scalable mental health interventions for home-based seniors with chronic physical conditions before segueing to the next paragraph.

2. Include one paragraph in the Background summarizing how cognitive-behavioral theories and components (e.g., behavioral activation, cognitive restructuring, etc.) that were entirely self-guided or augmented by a lay coach could help this population.

3. Lines 79-80: How many participants started therapy after enrolment?

4. Lines 101-103: What measures were the authors referring to when stratifying participants based on depression and/or anxiety scores? If they were PHQ-9 and GAD-7, the authors must list them from the outset.

5. Lines 113-121: Consistently spell out each questionnaire in full and then include the acronyms in parentheses. Do this for the GAD-7 and CAGE questionnaires.

6. Lines 113-142: Expand on this section by including the psychometric properties (i.e., internal consistency, retest reliability, and construct validity) of all study questionnaires.

7. Line 178: Briefly provide more information on the formal education the lay coaches received (e.g., undergraduate students, Bachelor’s degree-level coaches, etc.).

8. Lines 206-222: Instead of using the intent-to-treat analysis as a sensitivity analysis, make them the primary findings in the current study.

9. Lines 224-226: Briefly define what “stage of use” and “level of use” means by providing a few concrete examples for the reader. In addition, provide specific examples of primary and secondary tools.

10. Lines 207-214: Perform hierarchical linear modeling and determine if the pattern of results remains similar or not (Gallop & Tasca, 2009; Nezlek, 2012).

11. Lines 229-231: Conduct a power analysis (Arend & Schafer, 2019; Magnusson, 2018).

12. Lines 229-231: Detail how the authors managed the preprocessing steps (e.g., detecting and handling outliers).

13. Line 244: Briefly add how the researchers determined cognitive functioning levels.

14. Line 288: Do the authors mean between-group effect sizes? If so, please explicitly state that. In addition, include the effect sizes of time irrespective of the intervention group.

15. Lines 311-313: Explicitly spell out the measurement procedures to assess stage of use, level of use, and perceived helpfulness. What were the anchor points on these 3 or 4-point scales?

16. Lines 367-377: The absence of including quality of life measures is a limitation the authors should acknowledge.

17. The authors should advocate that future research conduct instrumental variable analyses to determine how adherence impacts outcomes in digital mental health intervention trials (Ten Have et al., 2008). They should also encourage future RCTs to determine if such interventions can affect cognitive functioning (Zainal & Newman, 2023).

18. Minor points:

a. Include a hyphen inside “CBT based” in the Abstract.

b. Line 15: Likewise, include a hyphen in “face to face.”

c. Line 33: Substitute “moved to evaluate” to “evaluated.”

d. Line 61: Remove the extra comma.

References

Arend, M. G., & Schafer, T. (2019). Statistical power in two-level models: A tutorial based on Monte Carlo simulation. Psychological Methods, 24(1), 1-19. https://doi.org/10.1037/met0000195

Gallop, R., & Tasca, G. A. (2009). Multilevel modeling of longitudinal data for psychotherapy researchers: II: The complexities. Psychotherapy Research, 19(4-5), 438-452. https://doi.org/10.1080/10503300902849475

Magnusson, K. (2018). powerlmm: Power analysis for longitudinal multilevel models. R package version 0.4.0. https://CRAN.R-project.org/package=powerlmm.

Nezlek, J. B. (2012). Multilevel modeling for psychologists. In APA handbook of research methods in psychology, Vol 3: Data analysis and research publication. (pp. 219-241). American Psychological Association. https://doi.org/10.1037/13621-011

Ten Have, T. R., Normand, S. L., Marcus, S. M., Brown, C. H., Lavori, P., & Duan, N. (2008). Intent-to-treat vs. non-intent-to-treat analyses under treatment non-adherence in mental health randomized trials. Psychiatric Annals, 38(12), 772-783. https://doi.org/10.3928/00485713-20081201-10

Zainal, N. H., & Newman, M. G. (2023). A randomized controlled trial of a 14-day mindfulness ecological momentary intervention (MEMI) for generalized anxiety disorder. European Psychiatry, 66(1), e12. https://doi.org/10.1192/j.eurpsy.2023.2

Reviewer #2: The current paper represents a randomized pilot trial comparing two brief self-care intervention strategies for the management of symptoms of depression and anxiety. As is the general theme of any pilot study should be, the current paper tries to address feasibility, and acceptability (explained in lines 47-54) and not the efficacy (or effectiveness) of any intervention. Since the does not plan to compare efficacy (or effectiveness) no power analysis is conducted and the sample size is based on convenience, which I accepted for a pilot study.

Two intervention arms are the “self-directed” group vs “coached group” and a CONSORT diagram is presented. Though initially nearly equal samples are allocated due to a higher number of “no longer interested”, coached group ended up with a smaller sample size (28 vs. 34). The authors may want to elucidate further why this difference and a potential reason for so.

The usage of the word “comparative effectiveness” in line 303 and elsewhere need additional explanations. For e.g., consider including “given this is a pilot RCT statistical significance (or lack of it) cannot be claimed, yet results are encouraging and the test for efficacy (or effectiveness) must be tested in an appropriately powered RCT”.

Overall, this is an interesting paper that needs minor modifications.

6. PLOS authors have the option to publish the peer review history of their article (what does this mean?). If published, this will include your full peer review and any attached files.

Reviewer #1: No

Reviewer #2: No

---

## [Author Response · Author response to Decision Letter 0]

20 Jun 2023

We thank the reviewers for their comments and revisions. References for location of changes (lines) in the responses below pertain to the version of the manuscript with tracked changes (reviewers’ reference to specific lines may no longer match up with the revised manuscript). Note that minor additional editorial changes were made throughout. We have also incorporated editable versions of the tables directly in the manuscript as per the editor’s request. 

Reviewer #1: 

1. Background, lines 2-8: The paragraph should end with a sentence highlighting the importance of identifying and developing scalable mental health interventions for home-based seniors with chronic physical conditions before segueing to the next paragraph. 

 We have added a transition sentence at the end of the first paragraph. 

2. Include one paragraph in the Background summarizing how cognitive-behavioral theories and components (e.g., behavioral activation, cognitive restructuring, etc.) that were entirely self-guided or augmented by a lay coach could help this population. 

 We have modified the second paragraph in the Background to include more information on the relevance of CBT interventions for the target population and included new references. 

3. Lines 79-80: How many participants started therapy after enrolment? 

 We have added the following information to the results section (see line 292): “One participant (in the control group) started therapy after enrolment.” 

4. Lines 101-103: What measures were the authors referring to when stratifying participants based on depression and/or anxiety scores? If they were PHQ-9 and GAD-7, the authors must list them from the outset. 

 We have specified which measures were used in the “Randomization” subsection, see lines 115-118. 

5. Lines 113-121: Consistently spell out each questionnaire in full and then include the acronyms in parentheses. Do this for the GAD-7 and CAGE questionnaires. 

 We have added the information as requested throughout the “Measures used, data collection” subsection. The CAGE questionnaire is now referred to as the 4-item alcohol abuse screening questionnaire (CAGE is derived from the four questions of the tool: Cut down, Annoyed, Guilty, and Eye-opener and is not an acronym per se). 

6. Lines 113-142: Expand on this section by including the psychometric properties (i.e., internal consistency, retest reliability, and construct validity) of all study questionnaires. 

 Additional information on psychometric properties of measures used added throughout the “Measures used” subsection. 

7. Line 178: Briefly provide more information on the formal education the lay coaches received (e.g., undergraduate students, Bachelor’s degree-level coaches, etc.). 

 Please note addition at line 204. 

8. Lines 206-222: Instead of using the intent-to-treat analysis as a sensitivity analysis, make them the primary findings in the current study. 

 The data analysis section, Table 4 and Appendix 2 have been updated.

9. Lines 224-226: Briefly define what “stage of use” and “level of use” means by providing a few concrete examples for the reader. In addition, provide specific examples of primary and secondary tools. 

 The stage and level of use measures are described on lines 165-172 as part of the “Measures user, data collection” sub-section of the Methods section. The primary and secondary tools are described in the ‘Intervention” sub-section (lines 196-201). 

10. Lines 207-214: Perform hierarchical linear modeling and determine if the pattern of results remains similar or not (Gallop & Tasca, 2009; Nezlek, 2012). 

 Since patient recruitment (level 1) was not conducted across different institutions or clusters (level 2), we do not see the need to apply multilevel modeling. If the reviewer is referring to multilevel modeling for longitudinal data, note that we do not have repeated measures (level 1) for each patient (level 2), but only 2 measures (Baseline and 2-month). Could the reviewer please elaborate more on the reasons for this request if additional analyses are required?

11. Lines 229-231: Conduct a power analysis (Arend & Schafer, 2019; Magnusson, 2018). 

 We did not perform a power analysis since it is a pilot study. Note that Reviewer 2 accepts that no power analysis is conducted as this is a pilot study.

12. Lines 229-231: Detail how the authors managed the preprocessing steps (e.g., detecting and handling outliers). 

 One new sentence was added to data analysis section (lines 249-250). 

13. Line 244: Briefly add how the researchers determined cognitive functioning levels. 

 We have specified that this was based on the BOMC scores from screening (see line 279) 

14. Line 288: Do the authors mean between-group effect sizes? If so, please explicitly state that. In addition, include the effect sizes of time irrespective of the intervention group. 

 The Effect size (Cohen d) is defined as the mean difference between the 2 study groups divided by the pooled standard deviation. A more detailed definition was added in the data analysis section. (Line 244-245)

15. Lines 311-313: Explicitly spell out the measurement procedures to assess stage of use, level of use, and perceived helpfulness. What were the anchor points on these 3 or 4-point scales? 

 Please see item 9 further above, note that we have repeated the detail regarding the scales for these measures in the discussion section as per the reviewer’s recommendation. 

16. Lines 367-377: The absence of including quality of life measures is a limitation the authors should acknowledge. 

 We have noted this limitation in the discussion as suggested (see lines 426-431). 

17. The authors should advocate that future research conduct instrumental variable analyses to determine how adherence impacts outcomes in digital mental health intervention trials (Ten Have et al., 2008). They should also encourage future RCTs to determine if such interventions can affect cognitive functioning (Zainal & Newman, 2023). 

 We have added such a recommendation to the conclusion (lines 452-453)

18. Minor points addressed: 

a. Include a hyphen inside “CBT based” in the Abstract.

b. Line 15: Likewise, include a hyphen in “face to face.”

c. Line 33: Substitute “moved to evaluate” to “evaluated.”

d. Line 61: Remove the extra comma.

Reviewer #2: 

The current paper represents a randomized pilot trial comparing two brief self-care intervention strategies for the management of symptoms of depression and anxiety. As is the general theme of any pilot study should be, the current paper tries to address feasibility, and acceptability (explained in lines 47-54) and not the efficacy (or effectiveness) of any intervention. Since the does not plan to compare efficacy (or effectiveness) no power analysis is conducted and the sample size is based on convenience, which I accepted for a pilot study.

Two intervention arms are the “self-directed” group vs “coached group” and a CONSORT diagram is presented. Though initially nearly equal samples are allocated due to a higher number of “no longer interested”, coached group ended up with a smaller sample size (28 vs. 34). The authors may want to elucidate further why this difference and a potential reason for so. 

 We have observed similar differences in previous studies (higher numbers of drop-outs in the coached arm vs the self-directed or usual care arms). We have attributed this difference to the fact that patients in the coached arm may reach a level of satisfaction related to the self-care tools or their use thereof and therefore withdraw from the study once their needs are met. We have included this potential explanation in lines 380-383. 

 We note in the manuscript that compared to the self-directed group, participants in the coached arm had a greater frequency of low scores on the PHQ-9 / GAD-7 (i.e., possibly less symptoms of anxiety and depression overall) (see lines 288-289). We can posit that some of these participants felt the study materials / calls were less relevant for them. However, because the differences in PHQ-9/GAD-7 scores between both groups were not significant, we did not retain this explanation. 

The usage of the word “comparative effectiveness” in line 303 and elsewhere need additional explanations. For e.g., consider including “given this is a pilot RCT statistical significance (or lack of it) cannot be claimed, yet results are encouraging and the test for efficacy (or effectiveness) must be tested in an appropriately powered RCT”. 

 We agree and have added this sentence in the first paragraph of the discussion section. (Lines 356-359)

---

## [Decision Letter · Decision Letter 1]

16 Jan 2024

Self-care interventions to assist family physicians with mental health care of older patients during the COVID-19 pandemic: Feasibility, acceptability, and outcomes of a pilot randomized controlled trial

PONE-D-23-02036R1

Dear Dr. Manon De Raad,

We’re pleased to inform you that your manuscript has been judged scientifically suitable for publication and will be formally accepted for publication once it meets all outstanding technical requirements.

Kind regards,

Mulinda Nyirenda

Academic Editor

PLOS ONE

Additional Editor Comments (optional):

Thank you for the revisions made to the manuscript.

Please note comments from reviews.

Reviewers' comments:

Reviewer's Responses to Questions

**Comments to the Author**

1. If the authors have adequately addressed your comments raised in a previous round of review and you feel that this manuscript is now acceptable for publication, you may indicate that here to bypass the “Comments to the Author” section, enter your conflict of interest statement in the “Confidential to Editor” section, and submit your "Accept" recommendation.

Reviewer #1: All comments have been addressed

Reviewer #3: (No Response)

2. Is the manuscript technically sound, and do the data support the conclusions?

Reviewer #1: Yes

Reviewer #3: Yes

3. Has the statistical analysis been performed appropriately and rigorously? 

Reviewer #1: Yes

Reviewer #3: Yes

4. Have the authors made all data underlying the findings in their manuscript fully available?

Reviewer #1: No

Reviewer #3: Yes

5. Is the manuscript presented in an intelligible fashion and written in standard English?

Reviewer #1: Yes

Reviewer #3: Yes

6. Review Comments to the Author

Reviewer #1: This reviewer thinks the authors did an excellent job with the revisions and only has one minor comment below.

Line 45: Replace the +/- with the word "or."

Congratulations.

Reviewer #3: Not sure why the sentence in line 181-182 was added and what it is saying. Usually, a participant would only start treatment after enrolling.

7. PLOS authors have the option to publish the peer review history of their article (what does this mean?). If published, this will include your full peer review and any attached files.

Reviewer #1: No

Reviewer #3: No

---

## [Editor Report · Acceptance letter]

6 Feb 2024

PONE-D-23-02036R1 

PLOS ONE

Dear Dr. de Raad, 

I'm pleased to inform you that your manuscript has been deemed suitable for publication in PLOS ONE. Congratulations! Your manuscript is now being handed over to our production team.

Kind regards, 

on behalf of

Dr. Mulinda Nyirenda 

Academic Editor

PLOS ONE